# Tobacco use among designated air pollution victims and its association with lung function and respiratory symptoms: a retrospective cross-sectional study

Kenji Kotaki,[1] Hideaki Senjyu,[1] Takako Tanaka,[1] Yudai Yano,[1] Naomi Miyamoto,[1] Tsuyoshi Nishinakagawa,[1] Yorihide Yanagita,[1] Masaharu Asai,[1] Ryo Kozu,[3] Mitsuru Tabusadani,[2] Terumitsu Sawai,[1] Sumihisa Honda[1]

[1]Department of Cardiopulmonary Rehabilitation Science, Nagasaki University Graduate School of Biomedical Sciences, Nagasaki, Japan
[2]Center for Industry, University and Government Cooperation, Nagasaki University, Nagasaki, Japan
[3]Department of Rehabilitation Medicine, Nagasaki University Hospital, Nagasaki, Japan

**Correspondence to**
Hideaki Senjyu;
senjyu@nagasaki-u.ac.jp

## ABSTRACT

**Objectives:** We sought to elucidate the long-term association of tobacco use and respiratory health in designated pollution victims with and without obstructive pulmonary defects.

**Design:** A retrospective cross-sectional study.

**Setting:** The register of pollution victims in Kurashiki, Japan.

**Participants:** 730 individuals over 65 years of age previously diagnosed with pollution-related respiratory disease. Patients were classified into four groups according to their smoking status and whether they had obstructive pulmonary disease. We then compared the prevalence of respiratory symptoms and lung function over time between groups.

**Primary outcome measures:** Spirometry was performed and a respiratory health questionnaire completed in the same season each year for up to 30 years.

**Results:** Rates of smoking and respiratory disease were high in our sample. Although respiratory function in non-smoking patients did not completely recover, the annual rate of change in lung function was within the normal range (p<0.01). However, smokers had worse lung function and were more likely to report more severe pulmonary symptoms (p<0.01).

**Conclusions:** Patients' respiratory function did not fully recover despite improved air quality. Our results suggest that, in the context of exposure to air pollution, tobacco use causes additional loss of lung function and exacerbates respiratory symptoms.

## Strengths and limitations of this study

- This is the first study to have analysed effects of smoking for respiratory function and respiratory symptoms in a group of designated pollution victims in Japan for a period up to 30 years.
- Tobacco use caused additional loss of lung function and exacerbated respiratory symptoms in those who have been exposed to air pollution.
- We did not have data on the quantity of cigarettes smoked, the duration of cigarette smoking or individual-level exposure to environmental pollutants.
- We were unable to account for possible confounding factors such as socioeconomic class and other lifestyle factors.

## INTRODUCTION

In many Asian nations, particularly China, air pollution is increasingly being recognised as an important emerging environmental and public health issue. A previous work has shown that levels of ambient air pollution are associated with an elevated incidence of respiratory diseases and reported the prevalence of respiratory symptoms,[1] particularly asthma, pulmonary emphysema and chronic bronchitis.[2] Atmospheric pollution caused by particulate matter less than 2.5 μm in diameter (PM 2.5) is reported to be particularly acute in China and in surrounding countries as a result of wind dispersal. This is a growing public health issue in the region, especially given that PM 2.5 is capable of penetrating deep into the lung tissue and precipitating a number of respiratory diseases.[3]

Japan was also severely affected during its period of accelerated economic growth in the wake of post-war reconstruction in the 1960s, with both air pollution and the prevalence of respiratory diseases rising markedly. In Kurashiki, a city located in the Okayama Prefecture, there were reports of pulmonary

toxicity owing to oxidant air pollutants rising by a factor of 1.73 among the city's inhabitants in the period leading up to 1970 as yearly average $SO_3$ levels increased. The majority of pollution-related deaths in this period was due to acute exacerbation of asthma.

Although patients do not commonly recover fully from pollution-related respiratory diseases, $FEV_1/FVC$ ratios (calculated by dividing $FEV_1$ by FVC) may return to within the normal range once air quality is improved.[4] As a result, Japan passed the Pollution-Related Health Damage Compensation Law in 1973 with the aim of identifying designated air pollution victims within affected areas for compensation. Patients with a diagnosis of bronchial asthma, chronic bronchitis or pulmonary emphysema and who had been resident in any designated air pollution zone for a specified period of time were legally recognised as pollution victims. These designated victims were fully reimbursed the costs of all related medical treatment by the state. However, patients' lung function, as measured using $FEV_1/FVC$ ratios, was not among the diagnostic criteria used to determine eligibility for compensation, as no agreed clinical definition for chronic obstructive pulmonary disease (COPD) existed at that time.

Tobacco smoke has been shown to contain high levels of PM 2.5, and is likely to be associated with many health issues in common with air pollution. Tobacco use has been shown to reduce lung function, as measured using $FEV_1/FVC$ ratios, and has also been identified as a cause of COPD and other tobacco-related diseases. The WHO Framework Convention on Tobacco Control was therefore adopted in 2003.[5] However, tobacco use remains a significant public health issue. According to a WHO report, more than 1.1 billion people globally are regular smokers.[6] Furthermore, Asia accounts for a third of the total world cigarette consumption, and there has been a local rise in air pollution.[7]

Both air pollution and tobacco smoke have been shown to reduce patients' $FEV_1/FVC$ ratios, to increase the risk of obstructive ventilatory defects and to exacerbate existing respiratory symptoms. Tanaka et al[4] have reported that when air quality is improved, the annual change in patients' $FEV_1$ can return to within the normal range. However, the association of ongoing tobacco use with respiratory function and pulmonary symptoms in patients with already reduced $FEV_1/FVC$ ratios resulting from bronchial hypersensitivity due to air pollution has not yet been elucidated.

The purpose of the present study, carried out among a population of confirmed pollution victims living in a designated pollution-affected area, was to determine the association between tobacco use and measures of respiratory function and prevalence of pulmonary symptoms for a period up to 30 years after an improvement in air quality. In particular, we sought to compare these associations in patients with and without diagnosed obstructive ventilatory defects resulting from exposure to air pollution.

## METHODS
### Study design and setting
The present study was embedded in a retrospective cross-sectional analysis of lung function and respiratory symptoms in designated victims of pollution-related illness in Kurashiki between their initial evaluation (from 1976 to 1988) and follow-up in 2009. Study subjects were drawn from the register of pollution victims in Kurashiki. All study participants met the following conditions, as determined by the Public Relief System of Kurashiki City, in accordance with the Pollution-Related Health Damage Special Measures Law (1969) and the Pollution-Related Health Damage Compensation Law (1973): (1) they had resided or were employed within a designated air pollution zone and (2) they were diagnosed with chronic bronchitis, asthma or emphysema by a medical practitioner.

In accordance with the Public Nuisance Countermeasures Law (1967), registered victims were entitled to various forms of compensation including a monthly consultation with a doctor, yearly assessments of respiratory symptoms consisting of a detailed questionnaire and spirometry tests. These patients received treatment with inhaled corticosteroids and long-acting β2 agonists and expectorants, which may have caused improvements in lung function.[8] However, this study did not collect detailed data regarding the treatment regimen, and we were therefore unable to evaluate the association between tobacco use and treatment.

Of the 419 203 total residents (204 958 male, 214 245 female), 3838 (0.9%) were officially classified as designated pollution victims in the period up to 1988. The records of these 3838 residents were reviewed in 2009 with the permission of the Kurashiki City Public Office (figure 1). Of the 1392 remaining 'officially designated victims of pollution-related illness', we screened 774 individuals aged ≥65 years for inclusion in the study and to focus the study on elderly people. The elderly people have a higher risk factor for the onset of COPD and show remarkably more respiratory symptoms of COPD, as we restricted this study to patients aged over 65 years . After excluding 44 records that did not have complete spirometry data, the resulting sample size was 730 individuals.

We classified patients into two groups according to whether they had an obstructive impairment by using the $FEV_1/FVC$ ratios recorded at their initial evaluation. Participants in each group were then further subdivided according to whether they had ever smoked or not (current smoker or ex-smoker or ever smoked)—determined using self-reporting at follow-up, but we could not identify the smoking period. This resulted in four comparison groups, labelled as group A (n=48, current smokers and ex-smokers diagnosed with an obstructive impairment reporting), group B (n=169, non-smokers diagnosed with an obstructive impairment), group C (n=119, smokers and ex-smokers without a diagnosed obstructive impairment) and group D (n=394, non-

**Figure 1** Flow chart showing the selection procedure for study subjects. The study included officially designated victims of pollution-related illness in Kurashiki aged 65 years or older in 2009, and for whom full data were available for statistical analysis. Ceased to experience symptoms: Officially designated victims who did renew the designation because of the disappearance of clinical symptoms.

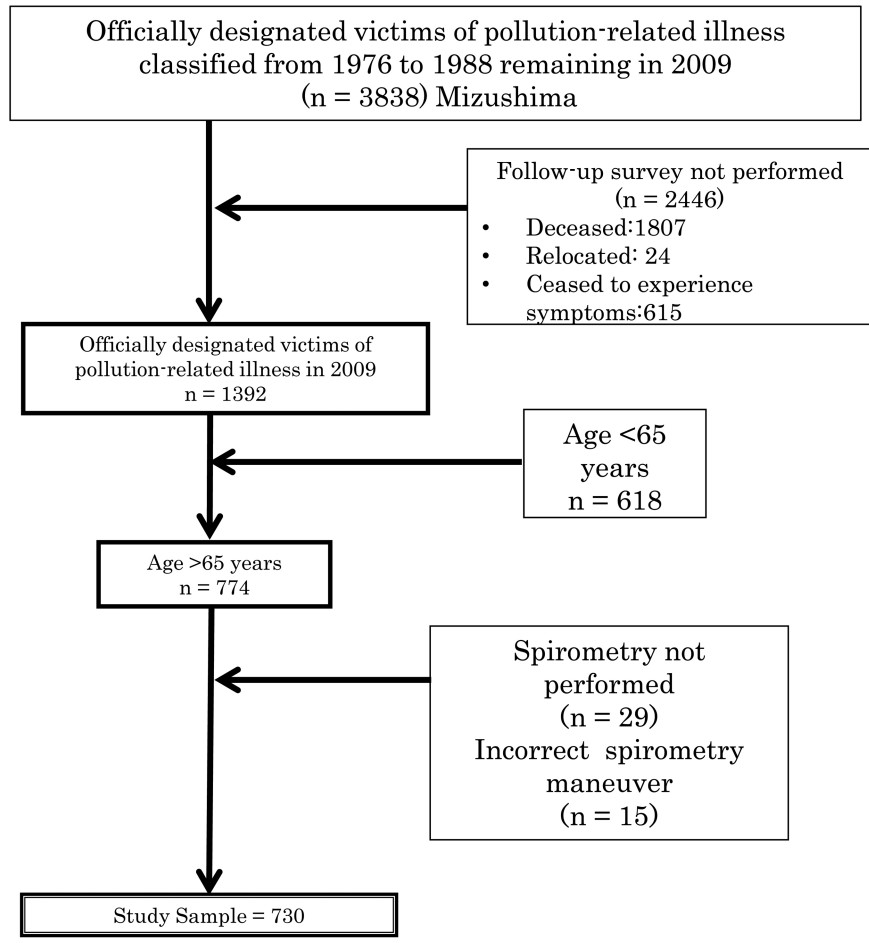

smokers without an obstructive impairment). The majority of these patients were diagnosed with chronic bronchitis (n=528, 68.2%), asthma (n=242, 31.3%) or emphysema (n=4, 0.5%), as described in case reports provided by authorised medical practitioners.

### Clinical measurements
Patients' gender, height, weight, body mass index, FVC, vital capacity (%VC), $FEV_1$, $FEV_1/FVC$ ratios, diagnosed respiratory conditions and respiratory symptoms were all recorded at baseline and at follow-up.

### Spirometry measurements
Spirometry was performed by trained staff at the Mizushima-Kyodo Hospital using an electronic spirometry device (FUDAC 70; Fukuda Sangyo Inc, Chiba, Japan). Tests were performed on patients in the sitting position, and were repeated until at least three reproducible forced expiratory curves had been obtained. Both $FEV_1$ and FVC were measured and used to calculate $FEV_1/FVC$ ratios in addition to %VC using the method proposed by Baldwin et al.[9]

### Annual change of spirometric data
The mean annual change in spirometry data was calculated by subtracting the follow-up results from the initial

evaluation results and dividing by the number of years between initial evaluation and follow-up.

### Respiratory symptoms
Respiratory symptoms (including dyspnoea, asthma attacks, cough and sputum production) were assessed by physicians in the same season each year using a respiratory health questionnaire designed by the government of Japan. Each symptom was graded using a standardised five-point scale.

Dyspnoea
1. Too breathless to leave the house, or breathless when dressing or undressing.
2. Breathless after walking about 50 m or after a few minutes on level ground.
3. Breathless when walking on level ground and keeping up with people of the same age, but not breathless when walking at own pace.
4. Breathless when walking up a slight hill or the stairs.
5. Breathless only during strenuous exercise.

Wheeze
1. Severe episode ≥10 days each month during the last year.
2. Severe episode ≥5 days each month during the last year, or mild episode ≥10 days each month during the last year.

3. Severe episode ≥1 day each month during the last year, or mild episode ≥5 days each month during the last year.
4. Mild episode ≥1 day each month during the last year.
5. No episodes of wheezing.

Cough and sputum
1. Daily cough and sputum, with a large amount of sputum or difficulty clearing sputum.
2. Daily cough and sputum, with a moderate amount of sputum or difficulty clearing sputum.
3. Daily cough and sputum, but not troublesome during daily life.
4. Daily cough and sputum for ≤3 months each year.
5. No cough or sputum.

### Air pollution monitoring
Data on mean daily concentrations of air pollutants were obtained at 21 points in Kurashiki. Monitoring of $SO_2$ concentrations started in 1965, while monitoring of nitrogen dioxide ($NO_2$) concentrations started in 1971.

### Statistical analysis
Data from patients in each of the four comparison groups were compared using the Kruskal-Wallis test and the Mann-Whitney U test as a post hoc test. Differences in gender distributions and the prevalence of respiratory diagnoses in each group were analysed using the Cochran-Armitage test, using the $\chi^2$ test to test for statistical significance. Changes in lung function between initial evaluation and follow-up were analysed using the Wilcoxon signed-rank test and the Mann-Whitney U test as a post hoc test. A normality test was performed on all continuous variables using the Kolmogorov-Smirnov test before analysis. All analyses were performed using PASW Statistics, V.18. Finally, p values of <0.05 were considered statistically significant.

### RESULTS
### Air pollutants
Figure 2 shows the annual mean daily levels of $SO_2$ and $NO_2$ recorded from 1965 to 2009 in Kurashiki. The Air Pollution Control Law was enacted in 1968. $SO_2$ levels were above the acceptable level every year from 1968 to 1974 and then decreased to below 40 parts per billion (ppb), which is the acceptable level defined by the Air Pollution Control Law. In 1973, the acceptable $NO_2$ level was changed by the Air Pollution Control Law from 20 to 40 ppb. The $NO_2$ level exceeded the revised acceptable level only in 1973.

### Sample characteristics
Table 1 shows the characteristics of our patient sample at initial evaluation. We found significant differences in age distributions between groups (p<0.001). Males outnumbered females in groups A and C (p<0.001). Patients in groups B and D had significantly lower height and weight (p<0.001).

We found differences in FVC values between groups, with patients in groups B and D presenting significantly lower values than those in groups A and C (p<0.001). $FEV_1$ values, which also differed significantly between comparison groups (p<0.001), were lowest in group B, followed by group A. %VC and FEV1/FVC ratios were lowest in groups A and B. Finally, the prevalence of bronchial asthma at initial evaluation was much higher in group B when compared with all other groups (p<0.001).

### Change in respiratory function and symptoms between initial evaluation and follow-up
Tables 2 and 3 show the results of our evaluation of the change in severity of respiratory symptoms and respiratory function between initial evaluation and follow-up. Respiratory function had deteriorated in almost all participants by follow-up. In particular, $FEV_1$/FVC ratios in group C, which averaged 79.1±5.6% at baseline, had fallen to 67.9±10.2% at follow-up because of the development of obstructive ventilatory defects in a number of participants. Conversely, average $FEV_1$/FVC ratios increased from 59.8±10.3% to 62.1±13.4% in group B (p<0.001). The median level of respiratory symptoms

**Figure 2** Sulfur dioxide and nitrogen dioxide concentrations from 1965 to 2009 relative to standard values.

**Table 1** Demographics and respiratory characteristics of comparison groups at initial evaluation for designation of air pollution victim status

| | Group A (n=48) | Group B (n=169) | Group C (n=119) | Group D (n=394) | p Value | Post hoc test |
|---|---|---|---|---|---|---|
| Age | 54.3±8.3 | 49.6±8.2 | 49.8±7.8 | 51.3±8.9 | <0.001 | b, c, f |
| Gender (male/female) | 35/13 | 66/103 | 81/38 | 90/304 | <0.001 | a, b, c, d, e, f |
| Height (cm) | 161.7±6.3 | 155.2±7.6 | 161.4±6.1 | 152.8±7.5 | <0.001 | a, b, d, e, f |
| Weight (kg) | 56.9±6.4 | 55.4±9.3 | 59.1±8.9 | 55.2±9.7 | <0.001 | b, f |
| BMI | 21.8±1.9 | 22.8±3.6 | 22.6±2.8 | 23.4±3.8 | <0.001 | a, b, d, e |
| FVC (L) | 3.37±0.79 | 3.04±0.79 | 3.49±0.76 | 2.94±0.68 | <0.001 | a, b, e, f |
| VC (%) | 105.5±18.7 | 103.5±18.5 | 108.3±15.5 | 107.6±17.3 | <0.05 | d, f |
| $FEV_1$ (L) | 2.09±0.64 | 1.72±0.58 | 2.67±0.58 | 2.24±0.58 | <0.001 | a, b, c, d, f |
| $FEV_1/FVC$ (%) | 64.9±6.5 | 59.8±10.3 | 79.1±5.6 | 79.6±7.1 | <0.001 | a, c, d, e, f |
| Data of duration after certification (years) | 25.9±4.1 | 26.9±4.4 | 24.8±4.2 | 25.5±4.6 | <0.001 | d, f |
| Diagnosed pulmonary disease CB, BA, PE CB+BA CB+PE, BA+PE (no) | 32/10/1/5/0/0 | 52/98/3/11/3/2 | 98/14/0/7/0/0 | 249/99/0/46/0/0 | <0.001 | a, c, d, f |
| Unit score | | | | | | |
| Dyspnoea | | | | | <0.001 | b, c, d, f |
| 1 | 0 (0.0%) | 1 (0.6%) | 0 (0.0%) | 0 (0.0%) | | |
| 2 | 0 (0.0%) | 4 (2.4%) | 0 (0.0%) | 4 (1.1%) | | |
| 3 | 9 (18.8%) | 31 (18.3%) | 7 (5.9%) | 55 (15.8%) | | |
| 4 | 30 (62.5%) | 111 (65.7%) | 75 (63.0%) | 246 (70.5%) | | |
| 5 | 9 (18.8%) | 22 (13.0%) | 37 (31.1%) | 89 (25.5%) | | |
| Asthma attack | | | | | <0.001 | a, b, d, f |
| 1 | 0 (0.0%) | 1 (0.6%) | 0 (0.0%) | 1 (0.3%) | | |
| 2 | 0 (0.0%) | 8 (4.7%) | 1 (0.8%) | 13 (3.3%) | | |
| 3 | 7 (14.6%) | 51 (30.2%) | 11 (9.2%) | 66 (16.8%) | | |
| 4 | 14 (29.2%) | 76 (45.0%) | 30 (25.2%) | 139 (35.3%) | | |
| 5 | 27 (56.3%) | 33 (19.5%) | 77 (64.7%) | 175 (44.4%) | | |
| Cough and phlegm | | | | | <0.05 | a, b, d, f |
| 1 | 0 (0.0%) | 2 (1.2%) | 0 (0.0%) | 0 (0.0%) | | |
| 2 | 0 (0.0%) | 5 (3.0%) | 1 (0.8%) | 9 (2.3%) | | |
| 3 | 24 (50.0%) | 50 (29.6%) | 58 (48.7%) | 143 (36.3%) | | |
| 4 | 23 (47.9%) | 90 (53.3%) | 58 (48.7%) | 225 (57.1%) | | |
| 5 | 1 (2.1%) | 22 (13.0%) | 1 (0.8%) | 15 (3.8%) | | |

Figures are presented as mean±SD.
Differences in sample characteristics analysed using the Kruskal-Wallis test with Cochran-Armitage tests for significance post hoc tests: Mann-Whitney U test, $\chi^2$ test for significance.
a: Group A vs group B.
b: Group C vs group D.
c: Group A vs group C.
d: Group B vs group D.
e: Group A vs group D.
f: Group B vs group C.
a, b: Current smokers/ex-smokers (groups A and C) vs never smokers (groups B and D).
c, d: Diagnosis of obstructive ventilatory defect (groups A and B) vs no obstructive ventilatory defect (groups C and D).
BA, bronchial asthma; BMI, body mass index; CB, chronic bronchitis; $FEV_1/FVC$ (%), forced expiratory volume in 1 s/forced vital capacity; PE, pulmonary emphysema; VC, vital capacity.

worsened from score 2 to 3 in all groups. However, group D saw a decrease in the proportion of patients reporting asthma attacks (p=0.78), cough and sputum production (p=0.24) over the period studied.

## Mean annual change in respiratory function

Table 4 shows the mean annual change in respiratory function in each comparison group.

Mean $FEV_1$ decreased to a similar extent in groups A and C. While there was a slight fall in $FEV_1$ in group D, there was a marginal improvement in $FEV_1$ in group B (p<0.001). At the same time, while $FEV_1/FVC$ ratios fell (p<0.001) in groups A and C, an improvement was seen in group B. Finally, %VC fell in all groups (p<0.001), although the decline was greater in groups A and B than in groups C and D, as shown in table 4.

**Table 2** Change in demographic characteristics and respiratory function of designated air pollution victims by comparison group between initial evaluation and follow-up

| | Initial evaluation | Follow-up | p Value | Initial evaluation | Follow-up | p Value | Initial evaluation | Follow-up | p Value | Initial evaluation | Follow-up | p Value |
| --- | --- | --- | --- | --- | --- | --- | --- | --- | --- | --- | --- | --- |
| | Group A | | | Group B | | | Group C | | | Group D | | |
| Age | 54.3±8.3 | 78.6±6.5 | <0.001 | 49.6±8.2 | 77.3±7.5 | <0.001 | 49.8±7.8 | 74.2±7.1 | <0.001 | 51.3±8.9 | 76.5±7.8 | <0.001 |
| Height (cm) | 161.7±6.3 | 159.3±6.9 | <0.001 | 155.2±7.6 | 150.8±8.8 | <0.001 | 161.4±6.1 | 159.2±6.2 | <0.001 | 152.8±7.5 | 148.9±8.8 | <0.001 |
| Weight (kg) | 56.9±6.4 | 55.4±9.2 | <0.001 | 55.4±9.3 | 52.1±10.9 | <0.001 | 59.1±8.9 | 58.9±9.6 | <0.001 | 55.2±9.7 | 53.2±11.3 | <0.001 |
| BMI | 21.8±1.9 | 21.7±2.9 | <0.001 | 22.8±3.6 | 22.7±3.9 | <0.001 | 22.6±2.8 | 23.2±3.1 | <0.05 | 23.4±3.8 | 23.9±4.2 | 0.13 |
| PFT | | | | | | | | | | | | |
| FVC (L) | 3.37±0.79 | 2.54±0.85 | <0.001 | 3.04±0.79 | 2.21±0.73 | <0.001 | 3.49±0.76 | 2.78±0.81 | <0.001 | 2.94±0.68 | 2.28±0.69 | <0.001 |
| VC (%) | 105.5±18.7 | 93.1±21.7 | <0.001 | 103.5±18.5 | 90.3±22.3 | <0.001 | 108.3±15.5 | 100.2±21.9 | <0.001 | 107±17.3 | 100.4±23.4 | <0.001 |
| $FEV_1$ (L) | 2.09±0.64 | 1.28±0.53 | <0.001 | 1.72±0.58 | 1.29±0.51 | <0.001 | 2.67±0.58 | 1.8±0.68 | <0.001 | 2.24±0.58 | 1.61±0.52 | <0.001 |
| $FEV_1$/FVC (%) | 64.9±6.5 | 54.5±2.3 | <0.001 | 59.8±10.3 | 62.1±13.4 | <0.001 | 79.1±5.6 | 67.9±10.2 | <0.001 | 79.6±7.1 | 73.8±13.8 | <0.001 |

Figures are presented as mean±SD.
Initial evaluation versus follow-up: Mann-Whitney U test.
PFT, pulmonary function testing; BMI, body mass index; $FEV_1$/FVC (%), forced expiratory volume in 1 s/forced vital capacity; VC, vital capacity.

**Table 3** Change in respiratory symptoms of designated air pollution victims by comparison group between initial evaluation and follow-up

| | Initial evaluation | | Follow-up | | p Value | Initial evaluation | | Follow-up | | p Value | Initial evaluation | | Follow-up | | p Value | Initial evaluation | | Follow-up | | p Value |
| --- | --- | --- | --- | --- | --- | --- | --- | --- | --- | --- | --- | --- | --- | --- | --- | --- | --- | --- | --- | --- |
| Unit score | Group A | | | | | Group B | | | | | Group C | | | | | Group D | | | | |
| Dyspnoea | | | | | | | | | | | | | | | | | | | | |
| 1 | 0 | 0.0% | 1 | 2.1% | <0.001 | 1 | 0.6% | 2 | 1.2% | <0.001 | 0 | 0.0% | 1 | 0.8% | <0.001 | 0 | 0.0% | 0 | 0.0% | <0.001 |
| 2 | 0 | 0.0% | 4 | 8.3% | | 4 | 2.4% | 15 | 8.9% | | 0 | 0.0% | 2 | 1.7% | | 4 | 1.1% | 13 | 3.7% | |
| 3 | 9 | 18.8% | 24 | 50.0% | | 31 | 18.3% | 74 | 43.8% | | 7 | 5.9% | 20 | 16.8% | | 55 | 15.8% | 67 | 19.2% | |
| 4 | 30 | 62.5% | 15 | 31.3% | | 111 | 65.7% | 75 | 44.4% | | 75 | 63.0% | 77 | 64.7% | | 246 | 70.5% | 279 | 79.9% | |
| 5 | 9 | 18.8% | 4 | 8.3% | | 22 | 13.0% | 3 | 1.8% | | 37 | 31.1% | 19 | 16.0% | | 89 | 25.5% | 35 | 10.0% | |
| Asthma attack | | | | | | | | | | | | | | | | | | | | |
| 1 | 0 | 0.0% | 0 | 0.0% | <0.001 | 1 | 0.6% | 0 | 0.0% | <0.001 | 0 | 0.0% | 0 | 0.0% | <0.001 | 1 | 0.3% | 0 | 0.0% | 0.78 |
| 2 | 0 | 0.0% | 3 | 6.3% | | 8 | 4.7% | 19 | 11.2% | | 1 | 0.8% | 4 | 3.4% | | 13 | 3.3% | 10 | 2.5% | |
| 3 | 7 | 14.6% | 16 | 33.3% | | 51 | 30.2% | 72 | 42.6% | | 11 | 9.2% | 17 | 14.3% | | 66 | 16.8% | 71 | 18.0% | |
| 4 | 14 | 29.2% | 15 | 31.3% | | 76 | 45.0% | 51 | 30.2% | | 30 | 25.2% | 35 | 29.4% | | 139 | 35.3% | 139 | 35.3% | |
| 5 | 27 | 56.3% | 14 | 29.2% | | 33 | 19.5% | 27 | 16.0% | | 77 | 64.7% | 63 | 52.9% | | 175 | 44.4% | 174 | 44.2% | |
| Cough and phlegm | | | | | | | | | | | | | | | | | | | | |
| 1 | 0 | 0.0% | 1 | 2.1% | <0.001 | 2 | 1.2% | 1 | 0.6% | <0.001 | 0 | 0.0% | 0 | 0.0% | <0.001 | 0 | 0.0% | 0 | 0.0% | 0.24 |
| 2 | 0 | 0.0% | 3 | 6.3% | | 5 | 3.0% | 8 | 4.7% | | 1 | 0.8% | 2 | 1.7% | | 9 | 2.3% | 9 | 2.3% | |
| 3 | 24 | 50.0% | 34 | 70.8% | | 50 | 29.6% | 99 | 58.6% | | 58 | 48.7% | 66 | 55.5% | | 143 | 36.3% | 155 | 39.3% | |
| 4 | 23 | 47.9% | 9 | 18.8% | | 90 | 53.3% | 55 | 32.5% | | 58 | 48.7% | 49 | 41.2% | | 227 | 57.1% | 221 | 56.1% | |
| 5 | 1 | 2.1% | 1 | 2.1% | | 22 | 13.0% | 6 | 3.6% | | 2 | 0.8% | 2 | 1.7% | | 15 | 3.8% | 9 | 2.3% | |

Numerical values are shown in %.
Initial evaluation versus follow-up: Wilcoxon signed-rank test.

**Table 4** Mean annual change in yearly spirometry results from initial evaluation to follow-up by comparison group

| | Group A (n=48) | Group B (n=169) | Group C (n=119) | Group D (n=394) | p Value | Post hoc test |
|---|---|---|---|---|---|---|
| $FEV_1$ decline/year (mL) | −36.7±19.1 | −26.5±18.5 | −37.2±16.4 | −25.5±17.2 | <0.001 | a, b, d, e, f |
| $FEV_1/FVC$ decline/year (%) | −0.04±0.05 | 0.09±0.06 | −0.04±0.03 | −0.02±0.04 | <0.001 | a, b, d, e, f |
| FVC decline/year (mL) | −31.7±24.5 | −30.1±20.7 | −29.9±22.8 | −25.3±17.1 | <0.05 | D |
| VC decline/year (%) | −0.05±0.08 | −0.05±0.08 | −0.03±0.06 | −0.03±0.07 | <0.001 | c, d, e, f |

Figures are presented as mean±SD.
p Values were calculated using the Kruskal-Wallis test post hoc test: Mann-Whitney U test.
a: Group A vs group B.
b: Group C vs group D.
c: Group A vs group C.
d: Group B vs group D.
e: Group A vs group D.
f: Group B vs group C.
a, b: Current smokers/ex-smokers (groups A and C) vs never smokers (groups B and D).
c, d: Diagnosis of obstructive ventilatory defect (groups A and B) vs no obstructive ventilatory defect (groups C and D).
$FEV_1/FVC$ (%), forced expiratory volume in 1 s/forced vital capacity; VC, vital capacity.

## Comparison of respiratory symptoms at final evaluation

Table 5 shows the severity of respiratory symptoms in our four comparison groups at follow-up.

In groups A and B, whose participants diagnosed with obstructive ventilatory defects at baseline, the great majority reported severe dyspnoea, asthma, cough and sputum production. Severe dyspnoea was also highly prevalent in groups A and B. While severity of asthma was high in groups B and D at initial evaluation, the proportion of patients reporting severe asthma had only increased significantly in groups A and C at follow-up (p<0.001). Severity of asthma was similar in groups C and D at follow-up. Prevalence of severe cough and sputum production increased significantly in groups A and C (p<0.001), with 80% of participants in group A reporting scores that changed from 1 to 3 due to a worsening of symptoms.

**Table 5** Change in severity of smoking-related respiratory symptoms by comparison group at follow-up

| Unit score | Group A (n=48) | | Group B (n=169) | | Group C (n=119) | | Group D (n=394) | | p Value | Post hoc test |
|---|---|---|---|---|---|---|---|---|---|---|
| Dyspnoea | | | | | | | | | | |
| 1 | 1 | 2.1% | 2 | 1.2% | 1 | 0.8% | 0 | 0.0% | <0.001 | c, d, e, f |
| 2 | 4 | 8.3% | 15 | 8.9% | 2 | 1.7% | 13 | 3.7% | | |
| 3 | 24 | 50.0% | 74 | 43.8% | 20 | 16.8% | 67 | 19.2% | | |
| 4 | 15 | 31.3% | 75 | 44.4% | 77 | 64.7% | 279 | 79.9% | | |
| 5 | 4 | 8.3% | 3 | 1.8% | 19 | 16.0% | 35 | 10.0% | | |
| Asthma attack | | | | | | | | | | |
| 1 | 0 | 0.0% | 0 | 0.0% | 0 | 0.0% | 0 | 0.0% | <0.001 | a, c, d, e, f |
| 2 | 3 | 6.3% | 19 | 11.2% | 4 | 3.4% | 10 | 2.5% | | |
| 3 | 16 | 33.3% | 72 | 42.6% | 17 | 14.3% | 71 | 18.0% | | |
| 4 | 15 | 31.3% | 51 | 30.2% | 35 | 29.4% | 139 | 35.3% | | |
| 5 | 14 | 29.2% | 27 | 16.0% | 63 | 52.9% | 174 | 44.2% | | |
| Asthma attack | | | | | | | | | | |
| 1 | 1 | 2.1% | 1 | 0.6% | 0 | 0.0% | 0 | 0.0% | <0.001 | a, b, c, d, e |
| 2 | 3 | 6.3% | 8 | 4.7% | 2 | 1.7% | 9 | 2.3% | | |
| 3 | 34 | 70.8% | 99 | 58.6% | 66 | 55.5% | 155 | 39.3% | | |
| 4 | 9 | 18.8% | 55 | 32.5% | 49 | 41.2% | 221 | 56.1% | | |
| 5 | 1 | 2.1% | 6 | 3.6% | 2 | 1.7% | 9 | 2.3% | | |

Numerical values are shown in %.
p values were calculated using the Kruskal-Wallis test post hoc test: Mann-Whitney U test.
a: Group A vs group B.
b: Group C vs group D.
c: Group A vs group C.
d: Group B vs group D.
e: Group A vs group D.
f: Group B vs group C.
a, b: Current smokers/ex-smokers (groups A and C) vs never smokers (groups B and D).
c, d: Diagnosis of obstructive ventilatory defect (groups A and B) vs no obstructive ventilatory defect (groups C and D).

## DISCUSSION

This retrospective cross-sectional study analysed changes in respiratory function and respiratory symptoms between two points in time (initial evaluation and follow-up) among a group of designated pollution victims in Japan. Our results show, for the first time, that improvements in respiratory function and reductions in respiratory symptoms are prevented by continuing tobacco use.

Among those patients diagnosed with an obstructive impairment due to air pollution exposure (groups A and B), we found that $FEV_1/FVC$ ratios had deteriorated by follow-up with no reduction in morbidity. However, our results showed an improvement in $FEV_1/FVC$ ratios in group B, which could be attributed to the high proportion of participants in this group diagnosed with bronchial asthma at initial evaluation when compared with the other groups.

We observed that respiratory function had deteriorated among patients in group A despite the improvement in local air quality, resulting in depressed $FEV_1/FVC$ ratios. Furthermore, while $FEV_1/FVC$ ratios in group C were relatively low at the time of initial diagnosis, continuing tobacco use had resulted in some cases in obstructive impairment, causing $FEV_1/FVC$ ratios at follow-up to decrease to $67.9\pm10.2\%$.

While air pollution can cause airway obstruction resulting in asthma, reactive airways dysfunction syndrome and reduced lung function, tobacco use, which can result in airway hyper-responsiveness, has been shown to exert a stronger influence on respiratory health in a number of studies.[10][11] These associations were evident in groups A and C, which showed similar annual rates of change in respiratory function over time. Conversely, Downs *et al*[12] have shown that, in non-smokers, annual rates of change in $FEV_1$ can return to within the normal range in response to an improvement in air quality. These findings are similar to the results of the present study. Previous studies have reported a mean annual rate of change in $FEV_1$ of between −30 and −22 mL/year in healthy, non-smoking males and females aged >65 years.[13] Patients with COPD had a mean annual change in $FEV_1$ of 30–80 mL/year.[14] However, exposure to air pollution and tobacco smoke, both of which can result in tracheobronchial injury, can worsen annual rates of change in lung function markedly.

A high proportion of our participants, particularly those with obstructive impairment, reported respiratory symptoms at baseline. These symptoms were more pronounced among the smokers in our sample. In group D, however, the proportion of patients reporting asthma attacks, sputum production and cough had not increased, suggesting that these symptoms may be prevented by smoking cessation. The proportion of participants in group B reporting asthma attacks had increased by follow-up. Furthermore, the proportion of participants reporting asthma attacks in group C had increased to the same levels as that found in group D at initial evaluation. Repeated bronchial inflammation caused by smoking and pollutant exposure can result in thickening of the bronchial wall and airway remodelling that cannot be reversed even by steroid treatment following the onset of hypersensitivity.[12] Furthermore, Siroux *et al*[15] have shown that continued tobacco use can significantly increase the incidence of asthma attacks. As a result of this association, smoking is listed as a potential trigger for asthma and COPD in asthma management guidelines.[16] Tobacco use is also likely to have been the cause of the higher prevalence of severe cough and sputum production found in groups A and C.

A number of studies have shown that smoking can cause respiratory tract inflammation and exacerbate symptoms such as cough and sputum production.[17] This was reflected in our finding that the severity of dyspnoea among participants in groups A and B had increased markedly at follow-up. While a number of causes of dyspnoea have been identified, including age, poor physical fitness, muscle atrophy and social and environmental factors, we could not identify the primary determinants of dyspnoea in the present study.[18]

Finally, the progression of respiratory disease and a decline in lung function can further aggravate these symptoms.[19]

In Asia, increased mortality and morbidity resulting from lung cancer, COPD, asthma and heart disorders caused by rising atmospheric PM 2.5 concentrations is a growing public health concern. According to the Organization for Economic Cooperation and Development, air pollution is expected to become the leading cause of death worldwide by 2050, and is likely to be responsible for between three and six million deaths annually.[20] Furthermore, Pope *et al*[21] have reported that worldwide pollution-related deaths could rise by 6% for each additional $10\,\mu g/m^3$ increase in atmospheric PM 2.5 concentrations.

The present study emphasises the need to prevent or reduce air pollution and to improve public education of the negative health effects of smoking, as well as to reduce pulmonary morbidity on the population level. This is especially the case among designated victims, for whom additional education programmes, behavioural interventions and support for smoking cessation would be particularly beneficial. One such behavioural intervention that has been proven to be cost-effective in reducing smoking prevalence is the five-step Global Initiative for Obstructive Lung Disease framework (ask, advise, assess, assist and arrange).[22] This strategy, which aims at delaying the progress of COPD through continuing support for smoking cessation, may be applicable for use in similarly affected communities.[23]

This study has several limitations. First, information on the number of cigarettes smoked and individual-level exposure to environmental pollutants was not available. Second, it was not within the scope of the present study to explore the role of gender-specific differences in exposure to tobacco smoke and pollutants, much of

which may have been occupational. Third, we were unable to clearly distinguish between the associations of mainstream and sidestream cigarette smoke when considering the associations of air pollution and health conditions. Finally, we were unable to account for possible confounding factors such as socioeconomic class and lifestyle factors, which may have resulted in individual-level differences in exposure to environmental pollutants.

Further work is needed to inform future prevention and treatment programmes to improve the quality of life of the remaining designated pollution victims in Japan. The results of the present study are of particular importance in the context of high rates of smoking and worsening air quality in a number of Asian countries as they undergo rapid economic change.

**Acknowledgements** The authors thank the study participants, technical staff, administrative support team and our co-workers for their help. We thank the Environmental Restoration and Conservation Agency and Mizushima-Kyodo Hospital for their support. In addition, the English language in this manuscript has been edited by the Edanz Group Ltd (Fukuoka, Japan).

**Contributors** HS was the principal investigator and contributed to the design of the study, administered the funding, supervised the team's work and made critical revisions to the manuscript for intellectual content. KK designed the study, collected, analysed and interpreted the data, and prepared the manuscript. SH took part in analysing and interpreting the data, and made critical revisions to the manuscript for intellectual content. All other authors took part in data collection and interpretation, and have read and approved the final manuscript.

**Funding** This work was supported by the Environmental Restoration and Conservation Agency.

**Competing interests** None.

**Ethics approval** The study protocol was approved by the Ethical Committee of the Nagasaki University Graduate School of Biomedical Sciences, 08072424-2.

**Provenance and peer review** Not commissioned; externally peer reviewed.

**Data sharing statement** Extra data can be accessed via the Dryad data repository at http://datadryad.org/ with the doi:10.5061/dryad.17tq7.

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
