## [Reviewer comments · BMJ Open]

Some articles will have been accepted based in part or entirely on reviews undertaken for other BMJ Group journals. These will be reproduced where possible.

ARTICLE DETAILS

TITLE (PROVISIONAL)	Tobacco use among designated air pollution victims and its association with lung function and respiratory symptoms: a retrospective cross-sectional study
AUTHORS	kotaki, kenji; Senjyu, Hideaki; Tanaka, Takako; Yano, Yudai; Miyamoto, Naomi; Nishinakagawa, Tsuyoshi; Yanagita, Yorihide; Asai, Masaharu; Kozu, Ryo; Tabusadani, Mitsuru; Sawai, Terumitsu; Honda, Sumihisa

VERSION 1 - REVIEW

REVIEWER	Hiroshi Mukae, M.d., Ph.D. Department of Respiratory Medicine University of Occupational and Environmental Health, Japan Japan
REVIEW RETURNED	16-Apr-2014

GENERAL COMMENTS	Kotaki and colleagues report the impact of smoking on designated air pollution-related pulmonary diseases. Though the impact of smoking on lung function is well known, the new insights of this report is the long-term impact of smoking on designated air pollution-related pulmonary diseases, and the data shown in this manuscript is precious for discussing health effect of long-term exposure to air-pollution. The strength of this study is that long-term effects of smoking in relation to air pollution exposure on spirometric data and respiratory symptoms in a certain area that air pollution had been severe. The weakness of this report may be incomplete information of smoking status as the authors describe as limitations in discussion. There are several points to be rearranged and rewritten in the manuscript described below. Major comments; 1. Other than smoking, air pollution level or used medication can also influence the annual change of lung function. I think you should also show the role of these effects.2. You showed that initial evaluation was from 1976-1988, and follow-up in 2009, in methods (Page 9). You also showed it was performed in 1976 and 2008 in discussion (Page 19).3. How did you estimate the annual change?4. Introduction is too long. You should show what you want to do clearly, simply and shortly in the introduction.
--

5. It is better to show more information of smoking status, for example, current or ex-smoking, and whether stopping or not during follow-up.

6. Comparing the data of this study to other data of annual decline in FEV1 and FVC in healthy nonsmokers, smokers, patients with COPD and asthma seems to be interesting.

7. There are many grammatical and spelling errors in the manuscript.

Minor comments;

1. In the title, if the victims only include air pollution victims, "designated pollution victims" may be better to be "designated air pollution victims".

2. Page 4, strength and limitation
Lines 5-7,

The victims of air pollution may be mainly because of environmental air pollution, and occupational exposure to ambient air pollution may not be considered.

3. Page 6, line 3
acute asthma
acute exacerbation of asthma?

4. Introduction
Please clearly separate discussions of smoking mainstream and sidestream cigarette smoke for considering health effect of air pollution.

5. Page 9, line 3
From 1976-1988
It is described as "1973 – 1988" in Figure 1.

6. Page 10, Paragraph 2, line 4-8,
Please describe each number in each group.

7. Page 11, Clinical measurements
Abbreviations of FVC, %VC, etc. are already shown in Introduction before.

8. Page 15, paragraph 1, line 5-6
Furthermore, patients in ...than...

9. Page 15, paragraph 2, line 1
We found differences in FVC values between groups..
-> at initial evaluation?

10. Page 20, paragraph 2 line 1
asthenia?

11. Page 21, Line 14
Reference 18 is not a guideline.

12. In Table 2
the numbers of patients in each pulmonary disease are hard to understand. In addition, please add the data of duration after

	certification. 13. Please reconstruct Table 3 to be easily understandable. 14. It is better to add the data of FVC in Table 4. 15. Figure 1 “Ceased to experience symptoms” Were the victims uncertified after disappearance of clinical symptoms? Please describe the reason why this study restricted patients over 65 years old.
--	--

REVIEWER	Bjorgulf Claussen Institute of Health and Society, University of Oslo, Norway
REVIEW RETURNED	29-Apr-2014

GENERAL COMMENTS	This is a well conducted and described study but I think the result, the importance of smoking on the Development by age of lung function and lung symptoms are not very interesting any more. This is a cross-sectional study in 2009 of a cohort which was defined by a new law as pollution victims in the city of Kurashiki, Japan, in 1973-88. They were examined annually from 1976 until the inclusion stopped in 1988 and again in 2009. The present article analyses the associations between lung function (FEV1/FEV ratio and self-reported lung symptoms) in 2009 in four groups, smokers/former smokers in 2009 and healthy/not healthy by inclusion in the sample, all for those 65 years of age or older. The authors show, not surprisingly, that smokers/former smokers have marked poorer lung function and more lung symptoms than non-smokers, and that those lung healthy at the start have a better lung function and less symptoms than those categorised as not lung healthy at the inclusion. This is the case in a city with steadily better air quality since 1973. The original sample is the 3,838 persons who were designated as pollution victims in 1976-88 but only 730 of them were included in the study in 2009. The reasons are somewhat differently explained in Figure 1 and at page 10 para 1. According to the figure, 1807 deceased, leaving 2031 alive (not 1,392 as stated in the text; to call them “survivors in 2009” in the figure is not correct). The figure tells us that 24 of those were relocated (moved away?), 615 ceased to experience symptoms (i.e. did not want to participate any more?), 618 were below age 65 and 44 did not have complete spirometric data. Thus, 19% of the original sample were followed up and analysed in 2009. The follow up time is 33 years, and considering that the sample per definition by law was unhealthy at the inclusion this is a good participation rate. However, a missing analysis must be done. The authors have data for all at start, and can compare the participants at follow up with those missing, especially the dead ones and those
--

	“without symptoms”. They were probably more or less unhealthy and maybe different on demographic variables at start. The authors do not explain why they exclude those under 65 years of age in 2009. They comprise 16% of the cohort. Were they examined in 2009? If included they would have increased the participation rate substantially. Otherwise I have no comments to the methods. But I think there are problems with the journalism here. Methods are complicated, and results are many. I spent too much time in grasping the content. Especially the names of the four groups were hard to remember, OS, ONS, NOS and NONS. Could you use words instead, for instance “unhealthy smokers”, “unhealthy non-smokers”, etc? The tables should be understandable without reading the text, and need longer headings. The signs for the posthoc tests are not properly defined in the footnotes. Most of the tests are significant, as far as I can see. Could this be simplified? I would skip Table 1, it is not used in the text except mentioned at page 9 para 1.
--	--

VERSION 1 – AUTHOR RESPONSE

1. . Reviewer’s comments:

Other than smoking, air pollution level or used medication can also influence the annual change of lung function. I think you should also show the role of these effects.

Response:

We have updated relevant sentences on the study design, setting, purpose, results, and subjects to address these effects (Page 6, lines 18–21; Page 10, line 5 -11). In addition, we have added Figure 2, showing the annual variation in levels of air pollutants.

2. . Reviewer’s comments:

You showed that initial evaluation was from 1976-1988, and follow-up in 2009, in methods (Page 9). You also showed it was performed in 1976 and 2008 in discussion (Page 19).

Response:

We have revised page 19 to eliminate the inaccurate reference. (Page 12, line 18)

3. Editor’s comments:

How did you estimate the annual change?

Response:

We have added an ‘Annual change’ subsection to the Methods section to clarify our procedure for estimating annual change. (Page 8, lines 6-10)

4. Editor’s comments:

The Introduction is too long. You should show what you want to do clearly, simply and shortly in the introduction.

Response:

We have revised the Introduction for clarity and length, as requested.

5. . Reviewer's comments:

It is better to show more information of smoking status, for example, current or ex-smoking, and whether stopping or not during follow-up.

Response:

We have revised the relevant sentence in the Methods section to clarify smoking status, as suggested. (Page 7, lines 7–9)

6. . Reviewer's comments:

Comparing the data of this study to other data of annual decline in FEV1 and FVC in healthy nonsmokers, smokers, patients with COPD and asthma seems to be interesting.

Response:

We have reworded the relevant sentences in the Discussion to discuss these relationships. (Page 13, lines 13–18)

7. . Reviewer's comments:

There are many grammatical and spelling errors in the manuscript.

Response:

The manuscript has been proofread and we have revised the manuscript accordingly to improve grammar and spelling throughout.

Minor comments;

1. Editor's comments:

In the title, if the victims only include air pollution victims, "designated pollution victims" may be better to be "designated air pollution victims"

Response:

We have changed the title accordingly.

2. . Reviewer's comments:

Page 4, strength and limitation

Lines 5-7,

The victims of air pollution may be mainly because of environmental air pollution, and occupational exposure to ambient air pollution may not be considered.

Response:

We have revised the relevant sentences in the strengths and limitations section. As you indicated, we had not considered occupational exposure, and the section has been rewritten to address this. (Page 2, lines 26–27)

3. . Reviewer's comments:

Page 6, line 3

acute asthma

acute exacerbation of asthma?

Response:

We have reworded this sentence. (Page 4, line 17)

4. . Reviewer's comments:

Introduction

Please clearly separate discussions of smoking mainstream and sidestream cigarette smoke for considering health effect of air pollution.

Response:

We were unable to clearly distinguish between the effects of mainstream smoke and side-stream cigarette smoke in this study, and we have updated the Discussion section to reflect this. (Page 15, lines 11–13)

5. . Reviewer's comments:

Page 9, line 3

From 1976-1988

It is described as "1973 – 1988" in Figure 1.

Response:

We have reworded the relevant sentences. (Page 6, line 7)

6. . Reviewer's comments:

Page 10, Paragraph 2, line 4-8,

Please describe each number in each group.

Response:

We have added sample sizes for all groups. (Page 7, lines 9–12)

7. . Reviewer's comments:

Page 11, Clinical measurements

Abbreviations of FVC, %VC, etc. are already shown in Introduction before.

Response:

We have deleted the redundant acronym definitions for FEV1 and FVC from page 7. lines 21-23

8. . Reviewer's comments:

Page 15, paragraph 1, line 5-6

Furthermore, patients in ...than...

Response:

We have revised page 10, lines 17-21 to clarify this sentence.

9. . Reviewer's comments:

Page 15, paragraph 2, line 1

We found differences in FVC values between groups..

-> at initial evaluation?

Response:

We have revised page 10, line 17 to clarify that the subsequent discussion describes the sample during initial evaluation.

10. . Reviewer's comments:

Page 20, paragraph 2 line 1

asthenia?

Response:

We have revised page 13, line 7. Asthenia was a misspelling of asthma.

11. . Reviewer's comments:

Page 21, Line 14

Reference 18 is not a guideline.

Response:

We have added reference 17, which is a guideline, to page 22, lines 4–5.

12. . Reviewer's comments:

In Table 2

the numbers of patients in each pulmonary disease are hard to understand. In addition, please add the data of duration after certification.

Response:

The Table 2 described here is now Table 1. We have revised Table 1 to clarify this material.

These totals have already been provided in the last row of table one in the second column listed as totals separated by / character. The row information of the table entitled "Diagnosed pulmonary disease" gives the key to understanding this information. CB: Chronic bronchitis etc. The key to reading this is listed after the table in the document.

13. . Reviewer's comments:

Please reconstruct Table 3 to be easily understandable.

Response:

We have divided Table 3 into two tables (Table 2 and Table 3) for clarity.

14. Editor's comments:

It is better to add the data of FVC in Table 4.

Response:

We have added the data of FVC in Table 4.

15. . Reviewer's comments:

Figure1"Ceased to experience symptoms"

Were the victims uncertified after disappearance of clinical symptoms?

Please describe the reason why this study restricted patients over 65 years old.

Response:

We have reworded the relevant sentences in the footnotes in Figure 1. This study was restricted to patients over 65 years old to focus the study on elderly people. The elderly have a higher risk of onset of COPD and show more respiratory symptoms of COPD which are remarkable, so we restricted the study to patients over 65 year of age.

A clarification has been added. (Page 6, line 27-Page7, line 2)

Dear Bjorgulf Claussen,

1. . Reviewer's comments:

This is a well conducted and described study but I think the result, the importance of smking on the Development by age of lung function and lung symptoms are not very interesting any more.

This is a cross-sectional study in 2009 of a cohort which was defined by a new law as pollution victims in the city of Kurashiki, Japan, in 1973-88. They were examined annually from 1976 until the inclusion stopped in 1988 and again in 2009. The present article analyses the associations between lung function (FEV1/FEV ratio and self-reported lung symptoms) in 2009 in four groups, smokers/former smokers in 2009 and healthy/not healthy by inclusion in the sample, all for those 65 years of age or older. The authors show, not surprisingly, that smokers/former smokers have marked poorer lung function and more lung symptoms than non-smokers, and that those lung healthy at the start have a better lung function and less symptoms than those categorised as not lung healthy at the inclusion. This is the case in a city with steadily better air quality since 1973.

Response:

Thank you very much for your feedback.

This is the first study to have analysed changes in respiratory function and respiratory symptoms in a group of designated pollution victims in Japan spanning over 30 years. Patients were classified by whether or not they had obstructive pulmonary disease. If they had obstructive pulmonary disease, it means they might have suffered from bronchial hypersensitivity. Many cigarettes are consumed in Asia particularly, while air pollution is not necessarily increasing. I hope I can be of some help to Asian victims.

2. . Reviewer's comments:

The original sample is the 3,838 persons who were designated as pollution victims in 1976-88 but only 730 of them were included in the study in 2009. The reasons are somewhat differently explained in Figure 1 and at page 10 para 1. According to the figure, 1807 deceased, leaving 2031 alive (not 1,392 as stated in the text; to call them "survivors in 2009" in the figure is not correct). The figure tells us that 24 of those were relocated (moved away?), 615 ceased to experience symptoms (i.e. did not want to participate any more?), 618 were below age 65 and 44 did not have complete spirometric data. Thus, 19% of the original sample were followed up and analysed in 2009.A2)

Response:

We have reconfirmed and reworded Figure 1. As you pointed out, the phrase 'number of survivors' is not correct. 'Officially designated victims of pollution-related illness' is correct. We have revised page 6, lines 26 to reflect this.

3. . Reviewer's comments:

The follow up time is 33 years, and considering that the sample per definition by law was unhealthy at the inclusion this is a good participation rate. However, a missing analysis must be done. The authors have data for all at start, and can compare the participants at follow up with those missing, especially the dead ones and those "without symptoms". They were probably more or less unhealthy and maybe different on demographic variables at start.

The authors do not explain why they exclude those under 65 years of age in 2009. They comprise 16% of the cohort. Were they examined in 2009? If included they would have increased the participation rate substantially.

Response:

This study restricted patients over 65 years old to focus the study on elderly people.

The elderly have a higher risk of onset of COPD and show more respiratory symptoms of COPD which are remarkable, so we restricted the study to patients over 65 year of age.

4. Otherwise I have no comments to the methods. But I think there are problems with the journalism here. Methods are complicated, and results are many. I spent too much time in grasping the content. Especially the names of the four groups were hard to remember, OS, ONS, NOS and NONS. Could you use words instead, for instance “unhealthy smokers”, “unhealthy non-smokers”, etc?

Response:

The names have been changed to group A, group B, group C, group D.

5. Editor’s comments:

The tables should be understandable without reading the text, and need longer headings.

Response:

We have reconfirmed and we have reworded the heading for each table.

6. The signs for the post-hoc tests are not properly defined in the footnotes. Most of the tests are significant, as far as I can see. Could this be simplified?

To simplify the post-hoc test column, the symbols have been changed to letters specifying comparisons between groups a, b, c, and d. The inclusion of a letter in the table indicates significance.

7. Reviewer’s comments:

I would skip Table 1, it is not used in the text except mentioned at page 9 para 1.

Response:

We have deleted the original Table 1.

VERSION 2 – REVIEW

REVIEWER	Hiroshi Mukae, M.D., Ph.D. Department of Respiratory Medicine, University of Occupational and Environmental Health, Japan, Japan
REVIEW RETURNED	06-Jun-2014

GENERAL COMMENTS	The authors have answered and rewritten their manuscript properly according to the reviewers’ comments. There are several minor comments described below. Minor comments: Page 30, Methods, Study Design and Setting, Paragraph 2, Line 3- ...a detailed questionnaire and spirometry tests, and prescriptions for expectorants and bronchodilators. These patients received treatment with bronchodilators such as inhaled corticosteroids and long-acting β2 agonists, which may have caused improvements in lung function. ->...a detailed questionnaire and spirometry tests. These patients received treatment with inhaled corticosteroids and long-acting β2 agonists and expectorants, which may have caused improvements in lung function. Page 6, Paragraph 3, Line 5- The elderly people have a higher risk factor for the onset of COPD
--

	and Show more respiratory symptoms of COPD are remarkable, as we was restricted to the patients over 65 years. -> The elderly people have a higher risk factor for the onset of COPD and show remarkably more respiratory symptoms of COPD, as we restricted to the patients over 65 years in this study. Page 4, Paragraph 1, Line 5 particulate matter less than 2.5 µm in diameter PM2.5 is ->particulate matter less than 2.5 µm in diameter (PM2.5) is Page 8 Annual change -> Annual change of spirometric data Page 13, Paragraph 2, Line 1- While air pollution can cause airway obstruction resulting in asthma, reactive airway dysfunction syndrome and... ->While air pollution can cause airway obstruction resulting in asthma, reactive airways dysfunction syndrome and... Page 13, Paragraph 2, Line 7- These findings are reflected in the results of the present study. -> These findings are similar to the results of the present study.
--	--

REVIEWER	Bjorgulf Claussen University of Oslo Department of Health and Society Norway
REVIEW RETURNED	12-Jun-2014

GENERAL COMMENTS	This is an original paper following up clinically a part of a population defined legally in 1973. Hence the epidemiological Methods must be primitive. As a general epidemiologist and GP I am not sure how interesting the results are but I think the article should be published, and it is nearly Clear for that now. The article is much clearer and more easy to read now than the first edition. I have only a few comments.  1. Page 5 lines 25-26. The logic in this sentence is not easy to grasp, a few more words would do. 2. Page 6 para 4 describes the sample, referring to Figure 1, and that is ok. But I guess that in the figure there is an error, "Ceased to experience symptoms: Officially designated victims who did not renew the designation because of disappearance of clinical symptoms"? I now understand that a missing analysis is not appropriate in the present complicated population and sample. 3. Page 9 para 3 and the tables: I spent time in understanding the last two columns in the tables 1, 4 and 5, the single p value (nearly always statistically significant) and the post-hoc test but I think I understand it now, and then the
--

	explanations are clear enough. 4. Results, discussion and conclusion should not at all refer figures that are given in the tables, only summing the figures up in words. Then the text will be much more readable.
--	--

VERSION 2 – AUTHOR RESPONSE

Dear Dr Hiroshi Mukae

We respectfully thank you for reviewing our manuscript.
 We have revised this sentence as per your instructions.
 I greatly appreciate your guidance and encouragement again.

Responses to the Reviewer's comments

1. Reviewer's comment:

Page 6 , Methods, Study Design and Setting, Paragraph 2, Line 37-40

...a detailed questionnaire and spirometry tests, and prescriptions for expectorants and bronchodilators. These patients received treatment with bronchodilators such as inhaled corticosteroids and long-acting β 2 agonists, which may have caused improvements in lung function.

->...a detailed questionnaire and spirometry tests. These patients received treatment with inhaled corticosteroids and long-acting β 2 agonists and expectorants, which may have caused improvements in lung function.

Response:

As suggested, we have revised the relevant sentence in the Methods section (Page 6, Methods, Study Design and Setting, Paragraph 2, Line 37-40).

2. Reviewer's comment:

Page 6, Paragraph 3, Line 56- Page 7, Paragraph 1, Line 2

The elderly people have a higher risk factor for the onset of COPD and Show more respiratory symptoms of COPD are remarkable, as we was restricted to the patients over 65 years.

->The elderly people have a higher risk factor for the onset of COPD and show remarkably more respiratory symptoms of COPD, as we restricted to the patients over 65 years in this study.

Response:

As suggested, we have revised the relevant sentence in the Methods section (Page 7, Paragraph 1, Line1-2).

3. Reviewer's comment:

Page 4, Paragraph 1, Line 17-18

particulate matter less than 2.5 μ m in diameter PM2.5 is

->particulate matter less than 2.5 μ m in diameter (PM2.5) is

Response:

As suggested, we have revised the relevant sentence in the Introduction section (Page 4, Paragraph 1, Line 17).

4. Reviewer's comment:

Page 8

Annual change

->Annual change of spirometric data

Response:

As suggested, we have revised the relevant sentence in the Methods section (Page 8, Paragraph 2, Line 15).

5. Reviewer's comment:

Page 13, Paragraph 2, Line 1-2

While air pollution can cause airway obstruction resulting in asthma, reactive airway dysfunction syndrome and...

->While air pollution can cause airway obstruction resulting in asthma, reactive airways dysfunction syndrome and...

Response:

As suggested, we have revised the relevant sentence in the Discussion section (Page 13, Paragraph 2, Line 1-2).

6. Reviewer's comment:

Page 13, Paragraph 2, Line 17-

These findings are reflected in the results of the present study.

-> These findings are similar to the results of the present study.

Response:

As suggested, we have revised the relevant sentence in the Discussion section (Page 13, Paragraph 2, Line 17).

Dear Bjorgulf Claussen

We respectfully thank you for reviewing our manuscript.

We have revised this sentence as per your instructions.

I greatly appreciate your guidance and encouragement again.

Responses to the Reviewer's comments

Page 5 lines 25-26. The logic in this sentence is not easy to grasp, a few more words would do.

Response:

We are sorry for the confusion. We have revised this sentence to read: "Furthermore, Asia accounts for a third of total world cigarette consumption, and there has been a local rise in air pollution.(Page 5 lines 25-26.)

2. Reviewer's comment:

Page 6 para 4 describes the sample, referring to Figure 1, and that is ok. But I guess that in the figure there is an error, "Ceased to experience symptoms: Officially designated victims who did not renew the designation because of disappearance of clinical symptoms"? I now understand that a missing analysis is not appropriate in the present complicated population and sample.

Response:

We have reworded Figure 1. It now reads: "Officially acknowledged victims of a pollution-related illness at the time of the end of the pollution authorization system (in 1988)."

3. Reviewer's comment:

Page 9 para 3 and the tables: I spent time in understanding the last two columns in the tables 1, 4 and 5, the single p value (nearly always statistically significant) and the post-hoc test but I think I understand it now, and then the explanations are clear enough.

Response:

We appreciate you pointing it out.

4. Reviewer's comment::

Results, discussion and conclusion should not at all refer figures that are given in the tables, only

summing the figures up in words. Then the text will be much more readable.

Response:

We have erased unnecessary overlapped sentences in tables and figures, to make the text simpler and more readable. Below are the parts that we have deleted.

Page 10, Paragraph 3, Line 34-35: "(the mean ages were 54.3 ± 8.3 , 49.6 ± 8.2 , 49.8 ± 7.8 and 51.3 ± 8.9 in the Group A–D respectively)."

Page 10, Paragraph 3, Line 44-45: "(3.04 ± 0.79 and 2.94 ± 0.68 versus 3.37 ± 0.79 and 3.49 ± 0.76 , $p<0.001$)."

Page 10, Paragraph 3, Line 48-50: "($1.72\pm 0.58\text{ml}$), ($2.09\pm 0.64\text{ml}$), (57.9%)"

Page 11, Paragraph 2, Line 34-43: All figures after each Group.

Page 12, Paragraph 2, Line 42: "(57.9%)."

Page 12, Paragraph 4, Line 48: "(54.5 ± 12.3